# Association of Sedentary Behavior with Brain Structure and Intelligence in Children with Overweight or Obesity: The ActiveBrains Project

**DOI:** 10.3390/jcm9041101

**Published:** 2020-04-12

**Authors:** Juan Pablo Zavala-Crichton, Irene Esteban-Cornejo, Patricio Solis-Urra, José Mora-Gonzalez, Cristina Cadenas-Sanchez, María Rodriguez-Ayllon, Jairo H. Migueles, Pablo Molina-Garcia, Juan Verdejo-Roman, Arthur F. Kramer, Charles H. Hillman, Kirk I. Erickson, Andrés Catena, Francisco B. Ortega

**Affiliations:** 1PROFITH “PROmoting FITness and Health Through Physical Activity” Research Group, Sport and Health University Research Institute (iMUDS), Department of Physical and Sports Education, Faculty of Sport Sciences, University of Granada, 18071 Granada, Spain; ireneesteban@ugr.es (I.E.-C.); patricio.solis.u@gmail.com (P.S.-U.); jmorag@ugr.es (J.M.-G.); cristina.cadenas.sanchez@gmail.com (C.C.-S.); maria_92_rg@hotmail.es (M.R.-A.); jairohm@ugr.es (J.H.M.); pablomolinag5@gmail.com (P.M.-G.); ortegaf@ugr.es (F.B.O.); 2Faculty of Education and Social Sciences, Universidad Andres Bello, Viña del Mar 2531015, Chile; 3IRyS Research Group, School of Physical Education, Pontificia Universidad Católica de Valparaiso, 2374631 Valparaiso, Chile; 4College of Health and Human Services, University of North Carolina at Charlotte, Charlotte, NC 28262, USA; 5MOVE-IT Research Group and Department of Physical Education, Faculty of Education Sciences University of Cádiz, 11519 Cádiz, Spain; 6Biomedical Research and Innovation Institute of Cádiz (INiBICA) Research Unit, Puerta del Mar University Hospital University of Cádiz, 11009 Cádiz, Spain; 7Department of Rehabilitation Sciences, KU Leuven – University of Leuven, 3000 Leuven, Belgium; 8Mind, Brain and Behavior Research Center (CIMCYC), University of Granada, 18071 Granada, Spain; j.verdejo@gmail.com; 9Laboratory of Cognitive and Computational Neuroscience (UCM-UPM), Center for Biomedical Technology (CTB), Pozuelo de Alarcón, 28223 Madrid, Spain; 10Department of Psychology, Northeastern University, Boston, MA 02115, USA; a.kramer@northeastern.edu (A.F.K.); c.hillman@northeastern.edu (C.H.H.); 11Beckman Institute, University of Illinois at Urbana-Champaign, Champaign, IL 61821, USA; 12Department of Physical Therapy, Movement & Rehabilitation Sciences, Northeastern University, Boston, MA 02115, USA; 13Brain Aging & Cognitive Health Lab, Department of Psychology, University of Pittsburgh, 3601 Sennott Square, Pittsburgh, PA 15218, USA; kiericks@pitt.edu; 14Department of Experimental Psychology, Mind, Brain and Behavior Research Centre (CIMCYC), University of Granada, 18011 Granada, Spain; acatenam@gmail.com

**Keywords:** sedentariness, TV viewing, magnetic resonance imaging, brain, cognition, childhood, obesity

## Abstract

We investigated the associations of different sedentary behaviors (SB) with gray matter volume and we tested whether SB related to gray matter volume is associated with intelligence. Methods: 99 children with overweight or obesity aged 8–11 years participated in this cross-sectional study. SB was measured using the Youth Activity Profile-Spain questionnaire. T1-weighted images were acquired with a 3.0 T Magnetom Tim Trio system. Intelligence was assessed with the Kaufman Brief Test. Whole-brain voxel-wise multiple regression models were used to test the associations of each SB with gray matter volume. Results: Watching TV was associated with lower gray matter volume in six brain regions (β ranging −0.314 to −0.489 and cluster size 106 to 323 voxels; *p* < 0.001), playing video games in three brain regions (β ranging −0.391 to −0.359, and cluster size 96 to 461 voxels; *p* < 0.001) and total sedentary time in two brain regions (β ranging −0.341 to −0.352, and cluster size 897 to 2455 voxels; *p* < 0.001). No brain regions showed a significant positive association (all *p* > 0.05). Two brain regions were related, or borderline related, to intelligence. Conclusions: SB could have the potential to negatively influence brain structure and, in turn, intelligence in children with overweight/obesity.

## 1. Introduction

Sedentary behavior refers to any waking behavior defined by energy expenditure ≤ 1.5 metabolic equivalents (METs) while in a sitting, reclining or lying posture [1]. Sedentary behavior represents approximately 60% of the awake time of children and an increase has been reported in the last years [2,3]. Nowadays, there is a rise in the availability of electronic forms of entertainment for children, including television, internet, mobile phones and video games, which has resulted in an increase in the time spent in sedentary behaviors and has become the most common leisure activity [2,4]. There is compelling evidence showing that increasing sedentary behaviors may impair physical, mental and cognitive health in children and adolescents [5,6]. Importantly, the influence of sedentary behaviors on cognitive outcomes may vary depending on the type of sedentary behavior. For example, educational sedentary behavior may positively influence cognitive outcomes, while non-educational sedentary behavior may impair cognitive-related outcomes (e.g., executive function, intelligence, academic performance) [7,8,9,10,11]. However, less is known about the influence of sedentary behavior on brain health and its relationship with intelligence across the lifespan. 

The structure of the brain is a result of synchronized processes that reflect an interaction between environmental and genetic factors that affect specific neural functions [12]. Sedentary behavior is one environmental factor that can influence brain structure and function [13]. In healthy children, there are only four studies examining the influence of sedentary behaviors on the brain, with one focused on brain connectivity [14] and the other three on brain structure [12,15,16]. For example, the longitudinal studies from Takeuchi et al. in normal-weight children showed that while more time watching TV was associated with greater regional gray matter volume (i.e., in the frontopolar and the medial prefrontal areas, and the hypothalamus/septum and the sensorimotor areas), frequent internet use was associated with a reduction in regional gray matter volume (i.e., the prefrontal areas, the anterior cingulate, the insula, and the temporal and occipital areas) [15,16]. As such, it is relevant to examine both overall and specific sedentary behaviors in relation to brain structure in children.

This is especially relevant in children with overweight/obesity because the increase in sedentary lifestyle often leads to weight gain or an inability to maintain weight loss and, in turn, may impair executive function, intelligence and academic performance during childhood [17,18,19,20]. In addition, recent evidence in adults suggests that the brain volume of individuals who are overweight or obese was 10 years more “aged” than that of their lean peers [21]. In fact, body mass index (BMI) has been negatively associated with gray matter volume [22,23]. Lastly, despite the importance of cardiorespiratory fitness on gray matter volume in normal-weight children [24,25] and in overweight/obese children [26], previous studies did not consider cardiorespiratory fitness when examining the influence of sedentary behaviors on the brain structure in children. The above-mentioned findings emphasize the relevance of examining the associations of sedentary behaviors with brain structure in the context of overweight or obesity in the early stages of life. To the best of our knowledge, no previous study has examined the influence of sedentary behaviors on brain structure in children with overweight/obesity; nor how sedentary behavior-related differences in brain volumes are associated with intelligence. 

Therefore, the objectives of the present study were (i) to examine the associations of different sedentary behaviors (i.e., watching TV, playing video games, and total sedentary time) with gray matter volume in children who are overweight or obese while adjusting for relevant confounders (e.g., cardiorespiratory fitness) using a whole-brain analytical approach; and (ii) to examine whether the gray matter regions associated with previous sedentary behaviors are related to intelligence.

## 2. Materials and Methods

### 2.1. Participants

Participants were part of the ActiveBrains project (http://profith.ugr.es/activebrains). ActiveBrains is a randomized controlled trial which aimed at examining the effects of an exercise program on brain, cognitive and academic outcomes, as well as on physical and mental health outcomes in children who are overweight or obese [27]. The initial sample included 110 children with overweight/obesity aged 8–11 years from the ActiveBrains project (categorized based on the World Obesity Federation cut-off points) [28]. The present cross-sectional analysis used the baseline data prior to group randomization and included a total of 99 children who had overweight or obesity (10.0 ± 1.1 years; 60.6% boys) with valid data on sedentary behavior, brain outcomes, and crystallized and fluid intelligence. The measurements were carried out from November 2014 to February 2016. Parents or legal guardians were informed of the study’s purpose and written informed parental consents were obtained. The ActiveBrains project was conducted in accordance with the Declaration of Helsinki and was approved by the Ethics Committee on Human Research (CEIH) of the University of Granada and was registered on ClinicalTrials.gov (identifier: NCT02295072).

### 2.2. Sedentary Behavior

Sedentary behavior was measured using the Youth Activity Profile-Spain (YAP-S) questionnaire, an adapted version of the original YAP. The original YAP was developed by the Physical Activity and Health Promotion lab at Iowa State University (www.physicalactivitylab.org) and validated against accelerometry in a series of studies by Saint-Maurice et al. [29,30]. Specifically, the correlation between measured and predicted minutes of total sedentary behaviors were high (r = 0.75, *p* < 0.001) [30]. The YAP questionnaire was translated (into Spanish) and back-translated (to test possible deviations from the English original version), as well as culturally adapted in collaboration with the original authors of the YAP (more details in http://profith.ugr.es/yap). 

The YAP-S is a self-administered 7-day recall questionnaire feasible for use in children. The YAP-S includes a section to assess sedentary behavior with 5 brief self-report items asking about time spent watching TV, playing video games, using the computer, using a cell phone, and also includes an overall sedentary-time item. Participants were asked about “how much time,” on average, they spent in four sedentary activities per day during the last week (i.e., watching TV, playing video games, using a computer and using a cell phone). Each question is scored using a scale that ranges from 1 to 5: (i) 0 min, (ii) less than 1 h, (iii) between 1 and 2 h, (iv) between 2 and 3 h and (v) more than 3 h. We excluded the items using a computer and using a mobile phone because 91% (1.54 ± 0.82) and 86% (1.71 ± 1.06) of the sample, respectively, was categorized as spending 0 min and less than 1 h on those behaviors. In addition, participants were asked about “how much time” they spent in total in sedentary behaviors in a “normal week” and the answers ranged from 1 = “almost none of free time sitting” to 5 = “almost all free time sitting.”

### 2.3. Magnetic Resonance Imaging (MRI) Acquisition and Processing

MRI evaluation was performed with a 3.0 Tesla Magnetom Tim Trio system (Siemens Medical Solutions, Erlangen, Germany) with a 32-channel head coil. Three-dimensional, high-resolution, T1-weighted images were collected using a magnetization-prepared rapid gradient-echo (MPRAGE) sequence. Defined parameters were: repetition time (TR) = 2300 ms, echo time (TE) = 3.1 ms, inversion time (TI) = 900 ms, flip angle =9°, field of view (FOV) = 256 × 256, acquisition matrix = 320 × 320, 208 slices, resolution = 0.8 × 0.8 × 0.8 mm, and scan duration of 6 min and 34 s. Imaging data were pre-processed using Statistical Parametric Mapping software (SPM12; Wellcome Department of Cognitive Neurology, London, UK) implemented in Matlab (The MathWorks, Inc, Natick, MA, USA). Imaging pre-processing steps included quality control and alignment, segmentation into tissues (i.e., gray and white matter tissues, and cerebrospinal fluid), creation of a customized template using Diffeomorphic Anatomical Registration Through Exponentiated Lie algebra (DARTEL), spatial normalization and spatial smoothing. Additional information about imaging pre-processing steps is discussed elsewhere [26].

### 2.4. Intelligence

Intelligence was evaluated by the Kaufman Brief Intelligence Test (K-BIT) which consists of two subtests, vocabulary and matrices. The K-BIT has demonstrated good construct validity and external validity (construct and concurrent validity) [31] Specifically, raw scores from the vocabulary subtest show a steady increase throughout early adulthood, peaking during middle adulthood, and gradually decreasing over the rest of the lifespan; this pattern of growth is consistent with theories of crystallized intelligence [32]. Raw scores from the matrices subtest peak in late adolescence/early adulthood and decrease steadily throughout the rest of adulthood, which is consistent with theories of fluid intelligence [32,33] As such, the vocabulary subtest estimates a crystallized intelligence score, which refers to the ability to use previously acquired knowledge, and the matrices subtest estimates a fluid intelligence score, which refers to the ability to solve novel reasoning problems [34]. The K-BIT was individually administered to each child by trained evaluators [31]. The variables included in the analyses were the age-specific percentile of crystallized score, fluid score and a composite intelligence score including both crystallized and fluid intelligence measures [35].

### 2.5. Covariates

The set of covariates included were sex, peak height velocity (PHV), parental education, BMI and cardiorespiratory fitness. Pubertal maturity status was determined with PHV and was obtained through the Moore et al. equation for boys and girls [36]; PHV offset was calculated by the difference between PHV and chronological age. Parents reported their educational level and answers were categorized as neither parent having a university degree, one parent having a university degree or both parents having a university degree. BMI was derived as weight in kilograms divided by height in meters squared (kg/m^2^). Cardiorespiratory fitness was assessed via the 20 m shuttle-run test, and maximal oxygen consumption (VO_2_max, mL/kg/min) was calculated via the Lèger equation [37]. We included cardiorespiratory fitness as a covariate because previous findings with the present sample indicated that cardiorespiratory fitness was associated with gray matter volume in several cortical and subcortical brain regions [26].

### 2.6. Statistical Analysis

Participants’ characteristics are shown as the mean and SD for continuous variables and percentages for categorical variables. Prior to data analyses, all variables were checked for normality of distribution using the Kolmogorov–Smirnov test, and no transformations were needed. In addition, we explored the association of several potential confounders with sedentary behavior variables by Pearson’s bivariate correlations using IBM SPSS (version 21 for Macintosh; *p* set at < 0.05). All covariates except BMI were correlated with at least one sedentary behavior variable (absolute r values ranging from 0.175 to 0.403, *p* < 0.1). Although BMI was not significantly correlated with any predictor variables, we included BMI in the models due to the characteristics of the study (i.e., children with overweight/obesity). Furthermore, we included sex as a covariate because differences in brain structure have been observed between girls and boys in the present and previous studies in children and adolescents [38,39].

Voxel-based morphometry analyses through whole-brain voxel-wise multiple regression models were done in SPM12 for the analyses of imaging data. The associations between sedentary behaviors (i.e., watching TV, video games and total sedentary time) and gray matter volume were tested in separate regressions using two models. Model 1 included adjustment for sex, PHV, parental education and BMI. Since in previous studies we have seen that cardiorespiratory fitness is an important correlate of grey matter in this sample [26], we ran an additional model 2 which included adjustment for model 1 plus cardiorespiratory fitness. Sensitivity analyses were also performed to confirm whether results showed in model 1 were maintained after adjusting for total physical activity assessed with YAP-S instead of cardiorespiratory fitness (Appendix A). To determine the spatial extent threshold, AlphaSim as implemented in Resting-State fMRI Analysis Toolkit toolbox (RESTplus) was used. Input parameters included a brain mask (i.e., 128190 voxels) and a cluster connection radius (i.e., 5 mm) considering the actual smoothness of data after model estimation. The statistical significance at voxel-level (threshold, *p* < 0.001 uncorrected), along with the cluster size were indicated in the results. Finally, the resulting cluster sizes were adjusted to account for the non-isotropic smoothness of structural images in accordance with Hayasaka [40].

We extracted the eigenvalues of the peak coordinates of each significant cluster that showed association with each sedentary behavior variable. To estimate the explained variance for each sedentary behavior variable in relation to brain regions, we performed separate regression models in SPSS including each sedentary behavior variable as a predictor and the eigenvalue as an outcome, adjusted for the covariates mentioned above. In addition, we performed linear regressions in SPSS to test the association between the sedentary behavior-related mean gray matter volumes and intelligence adjusted for sex, PHV offset, parental education university level (neither/one/both) and BMI in model 1, and further adjusted for cardiorespiratory fitness in model 2. We defined statistical significance as a Benjamini–Hochberg False Discovery Rate q less than 0.05 to correct for assessing multiple gray matter-intelligence regressions.

## 3. Results

### 3.1. Associations Between Sedentary Behaviors and Gray Matter Volume

Table 1 shows the characteristics of the whole study sample stratified by sex. Overall, boys had higher total gray matter volume than girls (*p* < 0.001). Likewise, boys spent more time watching TV (*p* = 0.033) and playing video games (*p* < 0.001) than girls.

Table 2 and Figure 1 display the brain regions showing negative associations between each sedentary behavior and gray matter volume. In model 1 (i.e., adjusting for sex, PHV, parental education and BMI), a greater amount of time watching TV was associated with lower gray matter volumes (*p* < 0.001, k = 77) in six clusters with β ranging from −0.314 to −0.489 and cluster sizes between 106 voxels and 323 voxels, specifically in the frontal regions (i.e., the middle frontal gyrus and the pars triangularis of the inferior frontal gyrus), the parietal regions (i.e., the inferior parietal gyrus and the postcentral gyrus), the occipital regions (i.e., the lingual gyrus) and the calcarine cortex. In model 2, after further adjustment for cardiorespiratory fitness, the same six clusters remained significant (*p* < 0.001, k = 75) with β ranging from −0.312 to −0.488 and cluster sizes between 99 voxels and 340 voxels.

In model 1, a greater amount of time spent playing video games was related to lower gray matter volumes (*p* < 0.001, k = 44) in three clusters with β ranging from −0.391 to −0.359 and cluster sizes between 96 voxels and 461 voxels, and this relationship was only observed in the temporal regions (i.e., the fusiform gyrus and the bilateral inferior temporal gyrus)**.** In model 2, only two regions remained significant (*p* < 0.001, k = 45), specifically the fusiform gyrus (β = −0.372, k = 277 voxels) and the left inferior temporal gyrus (β = −0.378, k = 229 voxels).

In model 1, a greater amount of total sedentary time was associated with lower gray matter volumes (*p* < 0.001, k = 62) in two clusters of the cerebellum, specifically the crus I (β = −0.341, cluster size = 897 voxels) and the crus II (β = −0.352, cluster size = 2455 voxels)**.** In model 2, after further adjustment for cardiorespiratory fitness, the associations disappeared. No brain region showed a statistically significant positive association between any sedentary behavioral variable and gray matter volume. All the analyses were also performed including total physical activity measured with YAP-S instead of cardiorespiratory fitness, and results were virtually maintained as in model 1 (Appendix A).

### 3.2. Associations Between Sedentary Behavior-Related Gray Matter Volume and Intelligence

Table 3 shows associations of sedentary behavior-related associations in gray matter with intelligence after controlling for potential confounders. Regarding the brain regions previously associated with sedentary behaviors, two of 10 regions were related or borderline related to intelligence indicators. Specifically, the inferior temporal gyrus was marginally related to the crystallized score (β = 0.191, *p* = 0.063 in model 1, and β = 0.178, *p* = 0.087 in model 2). The cerebellum crus II was positively associated with the crystallized score (β = 0.292, *p* = 0.003 in model 1, and β = 0.288, *p* = 0.005 in model 2) and with the composite score (β = 0.229, *p* = 0.020 in model 1, and β = 0.195, *p* = 0.055 in model 2). However, after correcting for multiple comparisons, only the cerebellum crus II remained significantly related to the crystallized intelligence score.

## 4. Discussion

The main findings of this study were that (i) different sedentary behaviors were negatively associated with gray matter volume in children with overweight or obesity, with watching TV being the behavior associated with the most regions. Specifically, watching TV was related to lower volumes of gray matter in the frontal, parietal and occipital regions; playing video games was associated with lower volumes of gray matter in the occipito-temporal regions; and total sedentary time was associated with lower volumes of gray matter in the cerebellum; (ii) the association of watching TV and gray matter volume was independent of cardiorespiratory fitness, whereas the associations of playing video games and total sedentary time with gray matter volume were partially confounded by cardiorespiratory fitness; and (iii) gray matter in the cerebellum was positively associated with intelligence (i.e., crystallized intelligence score and composite intelligence score). These results suggest that different sedentary behaviors (i.e., watching television, playing video games) and total sedentary time have the potential to influence brain structure and, in turn, intelligence, while some of the associations might be confounded by variation in cardiorespiratory fitness.

There are only two previous longitudinal studies examining the influence of sedentary behaviors (i.e., watching TV and internet use) on gray matter volume in children [15,16]. One study found that watching more TV was associated with greater gray matter volume in the frontopolar and the medial prefrontal areas in cross-sectional and longitudinal analyses, as well as positive associations in areas of the visual cortex using cross-sectional analyses and positive associations in the hypothalamus/septum and the sensorimotor areas in longitudinal analyses [15]. Conversely, the most recent study from Takeuchi et al. revealed that frequency of internet use was not associated with regional gray matter volume in cross-sectional analyses. However, the longitudinal analysis revealed that frequency of internet use was associated with gray matter reductions in widespread clusters, including the bilateral perisylvian areas, the cerebellum and the subcortical regions (e.g., the hippocampus, the amygdala or the basal ganglia; [16]).

In the present study, we found that, independently of the type of sedentary behavior, greater sedentariness was associated with less gray matter volume. In particular, watching TV was related to lower volumes of gray matter in the frontal (i.e., the middle frontal gyrus and the pars triangularis of the inferior frontal gyrus), parietal (i.e., the inferior parietal gyrus and the postcentral gyrus) and occipital regions (i.e., the lingual gyrus and the calcarine cortex); playing video games was associated with lower volumes of gray matter in the temporal regions (i.e., the fusiform gyrus and the bilateral inferior temporal gyrus); and a greater total sedentary time was associated with lower volumes of gray matter in the cerebellum (i.e., the cerebellum crus I and II). Interestingly, while Takeuchi et al. found both positive and negative associations between sedentariness and gray matter volume depending on the type of sedentary behavior (i.e., watching TV and internet use, respectively), we found a consistently inverse and region-specific association between sedentary behaviors (i.e., watching TV, playing video games and total sedentary time) and gray matter volume. Therefore, sedentary behavior may negatively associate with gray matter volume and each behavior may have regional specific associations in overweight/obese children. While watching TV was mainly related to the frontal and parietal regions, playing video games was related to the temporal regions; in addition, both behaviors were related to specific regions with the occipital cortex (i.e., watching TV with the calcarine cortex and the lingual gyrus, and playing video games with the fusiform gyrus). There are several potential reasons why watching TV and playing video games may be associated with different parts of the brain. For instance, while watching TV has been shown to be implicated in cortical and subcortical structures related to visual processes [15], playing video games has been related to cortical structures involved in sustained attention and verbal memory [41]. Of note, the identified associations of sedentariness and gray matter volume were different for each behavior, suggesting that specific sedentary behaviors may influence the structure of distinctive brain regions. However, future studies in children should include functional MRI while doing different sedentary behaviors to further contrast these hypotheses.

Another important finding is the role of cardiorespiratory fitness in the above-mentioned associations. The association between watching TV and gray matter volume was independent of cardiorespiratory fitness, whereas for playing video games one of the three brain regions was no longer significantly associated (i.e., the right inferior temporal gyrus), and for total sedentary time both regions (i.e., the cerebellum crus I and crus II) were not significantly associated after additional adjustment for cardiorespiratory fitness. This may be related to previous studies showing that cardiorespiratory fitness is positively associated with the structure of the preadolescent human brain [24,25,26,42]. For example, in a previous study with the present sample we found that higher cardiorespiratory fitness was related to greater gray matter volume in the frontal regions (i.e., the premotor cortex and the medial primary motor cortex), the subcortical nuclei (i.e., the hippocampus and the caudate), the temporal regions (i.e., the inferior temporal gyrus and the parahippocampal gyrus) and the calcarine cortex [26]. Specifically, the inferior temporal gyrus was negatively influenced by video games, as we found in the present study, and positively influenced by cardiorespiratory fitness [26], which further highlights the specific role of cardiorespiratory fitness with the association of playing video games and brain structure. In addition, three previous studies found that higher cardiorespiratory fitness was related to greater volume in the subcortical regions in children. Chaddock et al. found that children with higher cardiorespiratory fitness had greater volume of the hippocampus and the dorsal striatum of the basal ganglia (i.e., the caudate, the putamen and the globus pallidus) than less fit children [24,25]. Ortega et al. found that children with higher cardiorespiratory fitness had enlarged regions in the amygdala, the hippocampus, the bilateral putamen and the pallidum [42]. Therefore, cardiorespiratory fitness may play a role in the associations between sedentary behaviors and gray matter volume in children.

Interestingly, total sedentary time was selectively related to lower gray matter in the cerebellum and, in turn, the cerebellum was positively associated with crystallized intelligence, but not with fluid intelligence. First, this may not be a direct association (i.e., more total sedentary time), but an indirect influence (lower physical activity levels) affecting the key role that the cerebellum has in motor control and movement [43,44,45]. Second, the importance of the cerebellum for intelligence has been previously reported in some populations, and the specific association with crystallized, rather than fluid, intelligence may be supported by previous functional imaging studies showing cerebellar activation in relation to language, attention and mental imagery [46,47,48]. However, in the present study we showed structural, but not functional, brain correlates of sedentary behaviors, and all these speculations about links with intelligence should be viewed with caution. Thus, future studies should examine brain functional correlates of sedentary behaviors in children.

There is insufficient evidence to mention with certainty the mechanisms that might explain the associations between gray matter volumes and sedentary behavior. A decrease in dendritic spines due to a reduction in use-dependent plasticity may be one of such mechanisms. [13,16]. Most of the studies mention the possible mechanisms that influence brain structure from the path of being physically active, such as the release of growth factors (for example, BDNF, IGF1, VEGF) [17,49,50,51,52,53,54], synaptogenesis and neurogenesis [55], or increased blood flow in the brain promoting the development of new neurons [56]. These mechanisms would be expected to conversely occur while being sedentary instead of active. However, further studies are needed to elucidate the potential mechanisms underlying the association between sedentary behaviors and brain structure in children.

### Limitations and Strengths

The present study has some limitations. For example, because of its cross-sectional design, causality between sedentary behavior and brain volume cannot be determined. Further, our sample of children with overweight/obesity limits the generalization of our findings to the entire range of BMI distribution. Lastly, the YAP questionnaire did not provide information on some important sedentary behaviors, such as education (e.g., reading a book, studying, etc.). The strengths of this study include the relatively large sample of children with MRI, the inclusion of different sedentary behaviors and the adjustment for important confounding variables such as cardiorespiratory fitness and total physical activity.

## 5. Conclusions

Our results provide support for the negative association between different sedentary behaviors and gray matter volume among children with overweight/obesity. Specifically, watching TV was related to lower volumes of gray matter in the frontal, parietal and occipital regions. Playing video games was associated with lower volumes of gray matter in the temporal and occipital regions; and total sedentary time was associated with lower volumes of gray matter in the cerebellum and, in turn, with intelligence. While cardiorespiratory fitness had no confounding effect in the association of watching TV and gray matter volume, it plays a role in attenuating the negative associations for playing video games and total sedentary time. These results suggest that sedentary behaviors (i.e., watching television, playing video games and total sedentary time) could have the potential to negatively influence brain structure and, in turn, measures of intelligence. Cardiorespiratory fitness could play a role in this association; however, longitudinal and experimental studies are needed to confirm the relevance of decreasing the time spent engaged in sedentary behaviors in children for brain maturation.

## Figures and Tables

**Figure 1 jcm-09-01101-f001:**
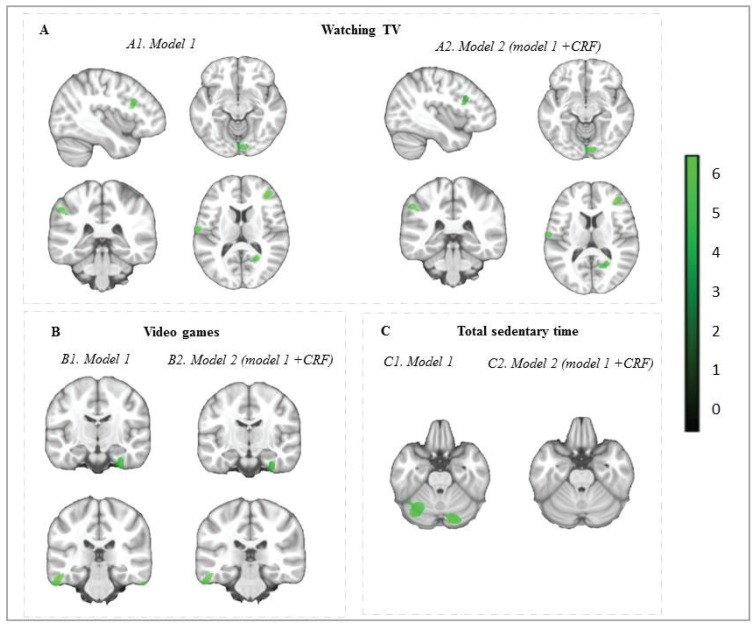
Brain regions showing negative separate associations of (**A**) watching TV, (**B**) video games and (**C**) total sedentary time with gray matter volume in children from the ActiveBrains projects. Analyses were adjusted for sex, peak height velocity offset (years), parent education university level (neither/one/both) and body mass index (kg/m^2^) in model 1, and model 2 was adjusted for model 1 plus cardiorespiratory fitness (CRF, mL/kg/min). Maps were thresholded using AlphaSim at *p* < 0.001with k = 77 voxels for model 1 and k = 75 for model 2 for watching TV, k = 44 voxels for model 1 and k = 45 for model 2 for video games, and k = 62 voxels for model 1 and k = 46 for model 2 for total sedentary time and surpassed the Hayasaka correction (see Table 2). The color bar represents T-values, with a lighter green color indicating a higher significant association. Images are displayed in neurological convention (i.e., the right hemisphere corresponds to the right side in coronal displays). Sagittal planes represent the left hemisphere. No brain regions showed a statistically significant positive association between any sedentary behavioral variable and gray matter volume.

**Table 1 jcm-09-01101-t001:** Characteristics of the study sample.

	All	Boys	Girls	*p* for Sex
*N*	99	60	39	
Physical characteristics				
Age (yr)	10.01 ± 1.14	10.16 ± 1.14	9.78 ± 1.12	0.108
Weight (kg)	55.87 ± 11.05	56.66 ± 10.69	54.65 ± 11.62	0.379
Height (cm)	143.91 ± 8.29	144.69 ± 7.37	142.72 ± 9.52	0.249
Peak height velocity offset (yr)	−2.31 ± 0.97	−2.65 ± 0.78	−1.78 ± 1.01	**<0.001**
Cardiorespiratory fitness (mL/kg/min) ^*^	40.77 ± 2.77	40.84 ± 2.77	40.65 ± 2.79	0.742
Body mass index (kg/m^2^)	26.77 ± 3.64	26.90 ± 3.79	26.57 ± 3.44	0.661
Body mass index category (%)				0.727
Overweight	25.3	26.7	23.1	
Obesity type I	43.4	45.0	41.0	
Obesity type II/III	31.3	28.3	35.9	
Parental education university level (%)				0.287
None of the parents	67.0	71.7	59.0	
One of the two parents	17.0	16.7	17.9	
Both parents	16.0	11.6	23.1	
K-BIT Intelligence score (1–99) **				
Crystallized score	56.37 ± 25.82	56.30 ± 25.89	56.49 ± 26.05	0.972
Fluid score	44.18 ± 25.64	40.62 ± 25.92	49.67 ± 24.52	0.086
Composite score	46.12 ± 25.16	43.65 ± 24.47	49.92 ± 26.05	0.227
Sedentary behaviors (1–5) ***				
Watching TV	2.92 ± 0.96	3.08 ± 1.05	2.67 ± 0.74	**0.033**
Playing video games	1.97 ± 1.07	2.32 ± 1.08	1.44 ± 0.82	**<0.001**
Total sedentary time	2.41 ± 1.13	2.35 ± 1.07	2.51 ± 1.23	0.488
Total gray matter (cm^3^)	793.56 ± 66.56	819.49 ± 56.13	753.67 ± 61.95	**<0.001**

Values are mean ± SD or percentages. * Measured with the 20-m shuttle run test. Lèger equation for transforming stage to VO^2^max (mL/kg/min). ** Measured with The Kaufman Brief Intelligence Test (K-BIT). *** Measured with Youth Activity Profile-Spain (YAP-S) questionnaire. Statistically significant values are shown in bold (*p* < 0.05)

**Table 2 jcm-09-01101-t002:** Brain regions showing negative associations between sedentary behaviors and gray matter volume in overweight/obese children (*n* = 99).

				Model 1				Model 2			
Brain Regions (mm^3^)	x	y	z	t	Cluster Size	B (95% CI)	β	t	Cluster Size	B (95% CI)	β
**Watching TV**											
Middle frontal gyrus	38	44	9	−5.45	228	−0.051 (−0.069–0.032)	−0.489	−5.47	231	−0.051 (−0.069–0.032)	−0.488
Inferior frontal gyrus, pars triangularis	−39	20	27	−4.03	127	−0.051 (−0.076–0.026)	−0.387	−4.03	132	−0.051 (−0.075–0.026)	−0.386
Inferior parietal gyrus	−53	−36	41	−3.89	105	−0.026 (−0.039–0.013)	−0.336	−3.87	102	−0.026 (−0.039–0.013)	−0.336
Lingual gyrus	11	−87	−11	−4.35	323	−0.025 (−0.036–0.013)	−0.378	−4.34	340	−0.024 (−0.035–0.013)	−0.382
Calcarine cortex	24	−59	12	−3.35	162	−0.031 (−0.049–0.013)	−0.314	−3.37	180	−0.031 (−0.049–0.013)	−0.312
Postcentral gyrus	−63	−12	11	−3.73	106	−0.022 (−0.033–0.010)	−0.357	−3.72	99	−0.021 (−0.033–0.010)	−0.356
**Playing video games**											
Fusiform gyrus	33	−14	−33	−4.95	461	−0.025 (−0.037–0.012)	−0.376	−3.88	277	−0.024 (−0.037–0.012)	−0.372
Inferior temporal gyrus	62	−24	−30	−3.52	96	−0.019 (−0.030–0.008)	−0.359	-	-	-	-
Inferior temporal gyrus	−54	−30	−21	−4.03	314	−0.030 (−0.045–0.015)	−0.391	−3.90	229	−0.029 (−0.044–0.014)	−0.378
**Total sedentary time**											
Cerebellum Crus I	20	−81	−27	−3.62	897	−0.025 (−0.039–0.011)	−0.341	-	-	-	-
Cerebellum Crus II	−42	−60	−42	−3.92	2455	−0.022 (−0.034–0.011)	−0.352	-	-	-	-

Analyses were adjusted for sex, peak height velocity offset (years), parent education university level (neither/one/both) and body mass index (kg/m^2^) for model 1. Model 2: Adjustments for model 1 plus cardiorespiratory fitness (mL/kg/min). All contrasts were thresholded using AlphaSim at *p* < 0.001with k = 77 voxels for model 1 and k = 75 for model 2 for watching TV, k = 44 voxels for model 1 and k = 45 for model 2 for playing video games, and k = 62 voxels for model 1 and k = 46 for model 2 for total sedentary time and surpassed the Hayasaka correction. Anatomical coordinates (x, y, z) are given in Montreal Neurological Institute (MNI) Atlas space. No brain regions showed a statistically significant positive association between any sedentary behavior variable and gray matter volume.

**Table 3 jcm-09-01101-t003:** Associations of sedentary behavior-related associations in gray matter with intelligence scores.

	Crystallized Score				Fluid Score				Composite Score			
	Model 1		Model 2		Model 1		Model 2		Model 1		Model 2	
	*β*	*p*	*β*	*p*	*β*	*p*	*β*	*p*	*β*	*p*	*β*	*p*
**Watching TV**												
Middle frontal gyrus	−0.072	0.455	−0.087	0.374	−0.085	0.395	−0.116	0.237	−0.087	0.359	−0.112	0.235
Inferior frontal gyrus, pars triangularis	0.069	0.470	0.058	0.544	0.050	0.611	−0.078	0.426	0.026	0.781	0.006	0.952
Inferior parietal gyrus	−0.105	0.326	−0.110	0.303	−0.030	0.787	−0.041	0.703	−0.077	0.466	−0.086	0.409
Lingua gyrus	−0.006	0.956	−0.017	0.871	−0.136	0.208	−0.163	0.124	−0.080	0.437	−0.101	0.322
Calcarine cortex	0.126	0.207	0.113	0.267	−0.013	0.903	−0.053	0.611	0.070	0.480	0.041	0.680
Postcentral gyrus	−0.004	0.966	−0.014	0.886	0.127	0.207	0.106	0.286	0.091	0.340	0.075	0.433
**Playing video games**											
Fusiform gyrus	0.029	0.783	0.021	0.841	−0.026	0.809	−0.045	0.676	0.013	0.900	−0.001	0.990
Inferior temporal gyrus	−0.054	0.591	−0.075	0.461	−0.044	0.673	−0.090	0.386	−0.040	0.684	−0.076	0.442
Inferior temporal gyrus	*0.191*	*0.063*	*0.178*	*0.087*	−0.007	0.947	−0.049	0.643	0.129	0.202	0.101	0.323
**Total sedentary time**											
Cerebellum Crus I	0.111	0.254	0.097	0.325	−0.036	0.720	−0.077	0.445	0.040	0.680	0.010	0.916
Cerebellum Crus II	**0.292**	**0.003 ***	**0.288**	**0.005 ***	0.140	0.179	0.084	0.434	**0.229**	**0.020**	*0.195*	*0.055*

Values are standardized regression coefficients (β). Model 1: Analyses were adjusted for sex, peak height velocity offset (years), parent education university level (neither/one/both) and body mass index (kg/m^2^). Model 2: Adjustments for model 1 plus and cardiorespiratory fitness (mL/kg/min). Statistically significant values are shown in bold (*p* < 0.05), and borderline significant values are shown in italics (*p* < 0.1). * These associations remained significant when P values were adjusted for multiple comparisons using the Benjamini and Hochberg method to control for the false discovery rate.

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
