# Peer review of "Association of Sedentary Behavior with Brain Structure and Intelligence in Children with Overweight or Obesity: The ActiveBrains Project"

_jcm, 2020, doi:10.3390/jcm9041101_

Round 1

Reviewer 1 Report

Main concern:

Introduction presents important topics related with the background of the study. I have particularly appreciate the second paragraph concerning the factors influencing the brain development. But, usually, the term "brain development" is reserved for brain formation before and immediately after birth. A second complementary process defined as "maturation", in terms of neuronal circuits up- or down-regulation (synapse formation or destruction), also represent an important point to eventually develop, because more related to the ages of children considered in the present study.

I have some reserves concerning the use of "intelligence": it appears lines 76, 87,92, and discussion and finally represent an important point of the whole study. Thus, the term "intelligence" is for some scientist not useable in modern argumentations. I suggest to define it with references or to replace eventually by "intellectual abilities", "cognitive abilities" or "problem resolution". Even in the cited references (17-20) the term "intelligence" appears only in one ref (20).

One more time in the method section line 103, it would be of great interest to define what is behind "intelligence variables".

At the end of the manuscript, lines 73-76 of the discussion, we find some arguments concerning specific neuronal functions that could be include in "intelligence". I suggest that this type of argumentation should be used more earlier in the text to fix what could be integrate in "intelligence" (or "intellectual abilities" or ....)

The following references could help:

Chen Y, et al. Sci Rep. 2019;9(1):2898

Nisbett RE, et al., Am Psychol. 2012;67(2):130-59.

Euler MJ. Neurosci Biobehav Rev. 2018;94:93-112

Minor remark:

To my mind, the argumentation concerning putative sex differences is weak, because presented within the "model1". I suggest to describe data in the light of parameters one by one before to focus on the two "models". It appears relevant since genders are presented in the method section; and because the data summary is also presented according genders in table 1. If no significant differences appeared between genders, or according each parameter one by one, please present it quickly before switching to the "models"

Author Response

Dear Reviewer:

Please see the atachment.

Best regards.

The authors.

Reviewer 2 Report

This cross-sectional study examined the association between questionnaire based sedentary behavior (SB), cardiorespiratory fitness, whole brain gray matter volume using VBM analysis and intelligence measured by the K-BIT in children with overweight or obese. The main findings are 1) watching TV was related to lower volumes of gray matter in the frontal, parietal and occipital regions; 2) playing videogames was associated with lower volumes of gray matter in the temporal and occipital regions; 3) total sedentary time was associated with lower volumes of gray matter in the cerebellum; 4) and there were no significant association between gray matter volume in these brain regions and intelligence, except the cerebellum. The research focus is interesting and significant in the modern society, however, the manuscript should more be improved especially on the following issues as below. 

Specific comments:

  1. According to description in the line 127-129 of pp3, total sedentary time was measured as rate in free time sitting, not exact spending time, though the questionnaire includes an overall sedentary time item (Saint-Maurice et al., 2015). Using overall sedentary time seems suitable, if the parameter is available.
  2. If I understand correctly, watching TV and playing videogame are part of total sedentary time. If so, it is not surprising if some TV or/and videogame related brain regions were associated with total sedentary time, but the results were not. Please discuss why do each SB associate with very different regions.
  3. The purpose and interpretation of the additional Model 2 analysis which adjusting for cardiorespiratory fitness is not clear. Considering CRF is a physiological landmark of habitual physical activity, it seems that significant association between SB and some brain regions even after adjusting for CRF means these brain regions are not associated with a greater amount of PA, but with watching TV and playing videogame itself. Please explain more about this point.
  4. In pp13 line 55, the authors discuss the possible association between cardiorespiratory fitness and gray matter volume in some brain regions (the right inferior temporal gyrus, the cerebellum crus 1 and crus 2). The authors should check this point with additional analysis.
  5. Although the authors hypothesized that “the influence of sedentary behaviors on cognitive outcomes may vary depending on the type of sedentary behavior. For example, educational sedentary behavior may positively influence cognitive outcomes…”, they did not measure educational sedentary behavior. This should be included in the limitations.
  6. Even spending time was very low (pp 3 line125), the sedentary activity time using the computer and using a mobile phone are interesting in the context of screen time. The data should be shown in the Table 1.
  7. In pp 12 line 35, the authors imply the present results are consistent with the findings of the previous studies of Takeuchi et al. (“Collectively, while Takeuchi et al. found both positive and negative associations between sedentariness and gray matter volume depending on the sedentary behavior (i.e., watching TV and internet use, respectively), we consistently found an inverse and region-specific association between sedentary behaviors (i.e., watching TV, playing video games and total sedentary time) and gray matter volume”). However, specific brain regions are dissociated each other, hard to consider “consist”.

Author Response

Dear Reviewer:

Best regards.

The authors.
